# Disease Reactivation in Secondary Progressive Multiple Sclerosis Patients Switching from Fingolimod to Siponimod: A Case Series

**DOI:** 10.3390/jcm11206033

**Published:** 2022-10-13

**Authors:** Gianmarco Abbadessa, Elisabetta Maida, Giuseppina Miele, Floriana Bile, Luigi Lavorgna, Simona Bonavita

**Affiliations:** Department of Advanced Medical and Surgical Sciences, University of Campania Luigi Vanvitelli, 80131 Naples, Italy

**Keywords:** secondary progressive multiple sclerosis, disease reactivation, disease modifying therapies, Sphingosine-1-phosphate modulator, fingolimod, Siponimod

## Abstract

Siponimod, a selective modulator of sphingosine 1-phosphate receptors 1 (S1P1) and 5 (S1P5), has recently been marketed for patients with Secondary Progressive Multiple Sclerosis (SPMS). Herein, we report three SPMS patients presenting disease reactivation in the first three months after switching from fingolimod to siponimod. Fingolimod binds to S1P1, S1P3, S1P4 and S1P5 receptors. S1P3 holds a central role in eliciting central proinflammatory responses, thus it has been hypothesized that upregulation of S1P3 may be the mechanism behind relapses after switching from fingolimod to siponimod. Further studies are needed to investigate the safety and efficacy of this treatment sequencing.

## 1. Introduction

Fingolimod was approved for the treatment of relapsing- remitting multiple sclerosis (RRMS) [1]. Several studies reported either disease reactivation or rebound after fingolimod discontinuation [2,3,4,5]. Although a proper definition has not yet been established, some authors defined rebound as the recurrence of disease activity exceeding the pre-treatment one; conversely, the recurrence of disease activity not exceeding the pre-treatment one is referred as a disease reactivation [6]. Although rebound and disease reactivation have been nearly exclusively observed in RRMS, few reports have described such condition after fingolimod discontinuation even in patients with Secondary Progressive Multiple Sclerosis (SPMS) [5,6].

Siponimod, a selective modulator of sphingosine 1-phosphate receptors 1 (S1P1) and 5 (S1P5), has recently been marketed for patients with SPMS [7]. We reported three SPMS patients presenting disease reactivation in the first three months after switching from fingolimod to siponimod.

## 2. Case Description

### 2.1. Case 1

A 47-year-old woman was diagnosed with RRMS at the age of 23. After INF beta 1a (44 μg three times weekly) treatment failure due to one relapse and the occurrence of 5 new T2 gadolinium-enhancing lesions, she was switched to fingolimod in July 2012. She experienced 2 relapses (September 2012 and January 2013) and then she was relapse-free till 2018, although an increase in brain and spinal cord MRI T2-lesion has been observed. From diagnosis to 2018, her EDSS progressed from 2,5 to 4. In April 2018, she experienced right retrobulbar optic neuritis, which receded after four days of high-dose corticosteroids. Since 2018 her disability gradually worsened, reaching in 2021 an EDSS of 6. In October 2021, she was switched to siponimod 2mg without a wash-out period. The MRI scan performed before siponimod initiation (September 2021) did not show gadolinium enhancement of lesions or new and/or enlarging T2 lesions compared to the previous MRI (September 2020). Twelve weeks after siponimod initiation, she experienced a relapse characterized by inguinal hypoesthesia and left lower limb weakness, needing constant bilateral assistance to walk (EDSS 6,5); occasional dysphagia for liquids also occurred. After five days of pulsed steroid therapy, she partially recovered from her new symptoms, again needing just a single aid to walk. Absolute lymphocyte count before starting siponimod was 550/µL and at three months follow up (just after the relapse) was 480/µL. Two months after remission, the patient performed brain, cervical and dorsal spinal cord MRI that showed an increased signal of some supratentorial lesions at DWI; however, it did not show either gadolinium enhancing lesions nor new and/or enlarging T2 lesions.

### 2.2. Case 2

A 36-year-old man was diagnosed with RRMS at the age of 23. After INF beta 1a (44 μg three times weekly) treatment failure due to the occurrence of 5 new T2 lesions, in December 2011, he was switched to fingolimod. He was free from disease activity (relapse, new and/or enlarging T2 lesions and/or gadolinium enhancing lesions) from 2011 to 2018, but EDSS increased from 1,5 to 2,5. In 2018 he relapsed, complaining of right lower limb hyposthenia; 5 days of pulsed steroid therapy induced partial recovery. Since 2018 his disability gradually worsened, reaching in 2021 an EDSS of 4. MRI performed immediately before siponimod initiation (October 2021) did not show gadolinium enhancement of lesions or new and/or enlarging T2 lesions compared to the previous scan (October 2020). In February 2022, he switched to Siponimod 1mg (CYP2C9*1*3 carrier) without a wash-out period. Forty-five days after siponimod initiation, he developed profound right lower limb hyposthenia with the inability to walk more than fifty meters (EDSS 6). After five days of pulsed steroid therapy, he had complete remission of symptoms. The absolute lymphocyte count before starting siponimod treatment was 574/µL; after the relapse, the total lymphocyte count was 790/µL. MRI scan was not performed.

### 2.3. Case 3

A 46-year-old woman was diagnosed with RRMS at the age of 25. After INF beta 1a (44 μg three times weekly) treatment failure due to one relapse and the occurrence of 1 gadolinium enhancing lesion and multiple enlarging T2 lesions, she was switched to fingolimod in July 2012. During the 3-year treatment with fingolimod she experienced 4 relapses (June 2013, March 2014, October 2014 and April 2015) and progressive increase in brain and spinal cord T2 lesion number. EDSS progressed from 2,5 to 4. She was, therefore, switched to Natalizumab with good control of disease activity for two years (she did not experience neither relapses nor MRI activity). In 2017, she was switched again to fingolimod for the risk of Progressive Multifocal Leukoencephalopathy (PML) due to the presence of high John Cunningham Virus (JCV) index (3.2). From 2017 to 2021, she experienced 3 clinical relapses (January 2018, May 2018 and August 2021) and her EDSS progressed to 6,5. MRI before siponimod initiation (October 2021) did not show neither gadolinium enhancing nor lesions or new and/or enlarging T2 lesions compared to the previous MRI (October 2017). In November 2021, she started treatment with siponimod 2mg without a wash-out period. After 3 months she had a brainstem relapse (vertigo and dysphagia) with only partial remission after 5 days of pulsed steroid therapy. Absolute lymphocyte count before siponimod initiation was 727/µL; after 3 months (after the relapse) it was 620/µL. MRI scan was not performed.

## 3. Discussion

Disease reactivation and rebound following fingolimod discontinuation have been extensively described [2,3,4,5]. However, it is remarkable that the three patients described experienced disease reactivation despite switching from fingolimod to another S1P modulator, without a wash-out period.

In our patients, we did not observe a higher disease activity after switching to siponimod than before fingolimod initiation. Therefore, a diagnosis of disease reactivation rather than rebound was proposed.

Recently, Senzaki and colleagues described a case of disease reactivation in a SPMS patient switching from fingolimod to siponimod [6]. Several studies reported that fingolimod discontinuation can cause reactivation of MS disease activity such as rebound or disease reactivation [2,3,4,5]. In the report of Evangelopoulos and colleagues [4], 3 out of 29 RRMS patients who stopped fingolimod, experienced a relapse (11%). Other cohort studies reported a similar frequency (San Francisco: 5/46 (10%) [2], Istanbul: 4/28 (14%) [3]. Almost all the exacerbations occurred between the 4th and the 16th week after fingolimod discontinuation, and this is consistent with fingolimod half-life (6–9 days) [2]. These retrospective observational studies highlight the possible risk of rebound and disease reactivation after fingolimod discontinuation, which appears to occur between the 4th and the 16th week and in about 10–15% of patients.

Since siponimod was available for the treatment of SPMS, 13 SPMS patients were switched from fingolimod to siponimod in our outpatient clinic. Among them, nine were switched for disease progression and four due to clinical relapse or MRI activity. Three patients out of 13 presented disease reactivations (23%). In all three patients, relapses occurred between 4 and 16 weeks after Fingolimod cessation, and this is consistent with the timing of disease reactivation due to fingolimod discontinuation described in the literature. Evidence suggests that overexpression of S1P receptors in lymphocytes occurs following fingolimod discontinuation. While the underlying mechanism is not yet entirely clear, this receptor overexpression could be driven by a quick repopulation of immune cells after fingolimod discontinuation, as also suggested by preclinical studies. Indeed, in the mouse model of experimental autoimmune encephalomyelitis (EAE) fingolimod discontinuation is followed by S1P1 overexpression and increase in Akt signalling in lymphocytes. This could enhance autoreactive lymphocyte ability to exit lymph nodes, thus favoring the encephalitogenic response. However, this mechanism would not be expected in these patients continuously treated with S1P1 modulators, as also suggested by the observed lymphocyte kinetics [8].

Behind the effect of immune cell trafficking, fingolimod could even modulate the inflammatory activity of resident immune cell of central nervous system. In support of a central immunomodulatory role exerted by fingolimod, studies in EAE showed that S1P stimulation (specifically S1P1 and S1P3) enhanced astrocytes proliferation and led to the release of nitric oxide and the translocation of nuclear factor-κB (NFκB) within astrocytes [9].

Further, Giordana and colleagues [10] found overexpression of S1P1 on astrocytes in the post-mortem brain of a patient who died of a fatal rebound after fingolimod discontinuation. They hypothesized that fingolimod rebound could be favored by the increased expression of S1P1 on astrocytes and, thus, of the downstream inflammatory response activity.

Several pharmacokinetic and pharmacodynamic differences exist between fingolimod and Siponimod [11,12]. 

First, unlike fingolimod, siponimod does not need to be phosphorylated to bind S1P receptors; this passage accounts for fingolimod long elimination half-life (12–16 h), compared to siponimod one, which is nearly halved (2–4.5 h) with lymphocyte counts returning to basal levels within about a week after treatment discontinuation [11,12]. Beyond lymphocyte kinetics, these pharmacokinetic differences could affect their capability of central nervous system (CNS) penetration. Indeed, the CNS penetration and, therefore, S1PR modulator effect in the CNS could be influenced by their lipophilicity, half-life, transport mechanisms and CNS metabolism (phosphorylation, oxidation) [13]. Some studies in (EAE) mice revealed differences in the bioavailability of fingolimod and siponimod in the CNS. Indeed, at doses producing full lymphocyte depletion, CNS penetration was 3- to 4-fold greater with fingolimod than with Siponimod [14]. However, drug bioavailability in the CNS does not correspond to the free drug levels available to act on receptors.

Concerning the pharmacodynamic features of S1PR modulators, while fingolimod binds to four (S1P1, S1P3, S1P4 and S1P5) of the five S1P receptors, siponimod does not bind S1P3 and S1P4. When binding to S1P1, both drugs initially act as agonists, but this is quickly overcome by functional antagonism that leads to the internalization and degradation of the receptor [11]. In contrast, a study on Chinese hamster ovary (CHO) cells showed a complete internalization of S1PR3 with fingolimod but not with siponimod [11], in line with their receptor-binding profiles. Neither drug resulted in the internalization of S1PR5 [15].

Further, evidence indicates that once phosphorylated fingolimod first acts as potent S1PR1 agonist accounting for the transient cardiac effects (needing for first dose observation) that is quickly replaced by functional antagonism. It occurs because binding at S1PR1 leads to the irreversible internalization of the receptor [11]. Siponimod has high affinity for S1PR1 and S1PR5 and produces manageable cardiac effects at initiation. This profile is coherent with transient agonism and subsequent functional antagonism of S1PR1. However, the manageable cardiac effects of siponimod are primarily due to the different receptor selectivity (the function and distribution of S1P3 in the ventricle cardiac conduction system suggest a key role of this receptor in the risk of type I and type II atrioventricular block) [16].

Astrocytes express on their surface S1P3 as well as S1P1. S1P3 is central in eliciting pro-inflammatory responses through its capacity to activate RhoA and its upregulation in astrocytes [17]. For this reason, it has been extensively considered a potential therapeutic target to slow down the progression of CNS diseases. Thus, it has been hypothesized that upregulation of S1P3 may be the mechanism behind relapses after switching from fingolimod to siponimod [6], by triggering a pro-inflammatory cytokine cascade via activation of NFκB. However, we could not exclude that other pharmacokinetic and/or pharmacodynamic differences might have played a role in these cases of disease reactivation.

## 4. Conclusions

These reports highlighted that disease reactivation after switching from fingolimod to Siponimod could occur even with a switch without a washout period. Further studies are needed to investigate the safety and efficacy of this treatment sequencing and to better understand the molecular mechanism of disease reactivation and rebound after fingolimod discontinuation.

## Data Availability

The data that support the findings of this study are available from the corresponding author upon reasonable request.

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
