# Peer review of "Disease Reactivation in Secondary Progressive Multiple Sclerosis Patients Switching from Fingolimod to Siponimod: A Case Series"

_jcm, 2022, doi:10.3390/jcm11206033_

Round 1

Reviewer 1 Report

Dear authors 

The manuscript “Disease reactivation in Secondary Progressive Multiple Sclerosis patients switching from fingolimod to siponimod: a case series.” presents interesting data on disease reactivation in SPMS patients switching between two SPRM. However, in my opinion the manuscript has some flaws, which have to be improved prior to publication. I therefore suggest major revisions prior to publication. See in more detail below.

Thank you for your work.

Best regards

Major aspects

·         The first presented case with the “inguinal hypoesthesia and left lower limb hyposthenia” could also be caused by a new spinal lesion. Why didn’t you perform a spinal MRI in addition to the cerebral MRI?

·         In all three cases you describe a “treatment failure” under IFN, please be more precise and describe, how treatment failed in each one of the patients.

·         In all three cases you do not describe the duration between stopping fingolimod and starting siponimod. In the discussion you mention that there was no wash-out period, so you stopped fingolimod on one day and started siponimod on the next day?

·         It is important to mention that the term “rebound” has not been precisely defined in the literature. Most commonly it describes a level of disease activity after stopping of an immunotherapy which is higher than before initiation of this immunotherapy. Moreover, the terms disease reactivation and disease rebound should not be used interchangeably.

·         In the discussion, you fail to mention other differences between fingolimod and siponimod besides the receptor selectivity, such as partial agonism/antagonism on the S1PR, precise targeting of the S1PR and different pharmacokinetic characteristics, which presumably contribute to the differences between the various S1P receptor modulators (see for example “FTY720 (fingolimod) for relapsing multiple sclerosis,” Expert Rev Neurother, 2008, “Sphingosine 1-Phosphate Receptor Modulators for the Treatment of Multiple Sclerosis,” Neurotherapeutics, 2017, and “Sphingosine 1-phosphate Receptor Modulator Therapy for Multiple Sclerosis: Differential Downstream Receptor Signalling and Clinical Profile Effects,” Drugs, 2021).

Minor aspects

·         Page 1, line 34: “A 47-year-old woman was diagnosed with RRMS at the age of 23 years old”. Please omit the word “old”, it is not necessary.

·         Page 1, lines 37-38: “… although an increase in brain and spinal cord MRI T2-lesion 37 have been observed.” Please change “have” to “has”.

·         Page 1, lines 38 and 40: “Her EDSS progressed from 2,5 to 4.” “Since 2014 her disability gradually worsened, reaching in 2021 an EDSS of 6.” Those sentences are a bit contradictory, please adapt.

·         Page 2, line 44: “left lower limb hyposthenia”: I would advise to change “hyposthenia” to “weakness” throughout the manuscript.

·         Page 2, line 53: “A 36-year-old man was diagnosed with RRMS at the age of 23 years old.” Please omit the word “old”.

·         Page 2, lines 55-56: “He was disease activity (relapse, new T2 lesion and/or Gd enhancing lesion) free from 2011 to 2018…” Please change the sentence order to “he was free from disease activity (relapse, new T2 lesion and/or Gd enhancing lesion) from 2011 to 2018”. Moreover, you omit in this definition of disease activity the enlarging T2 lesions, which is a bit unusual. Were the enlarging T2 lesions not included in your definition of disease activity?

·         Page 2, lines 74-75: “In 2017, she was switched again to fingolimod for the presence of high JCV index (3.2).” To be precise this sentence is not entirely correct. She was not switched due to the high JCV index, but the associated PML risk. Moreover, please introduce all abbreviations in your manuscript, when they are first used.

·         Page 3, lines 101-102: “. Three patients out of 13 presented disease reactivations (20%)”. According to my calculations 3 out of 13 equals 23%.

·         Page 3, lines 116-119: “In support of a central immunomodulatory role exerted by fingolimod, studies in EAE showed that S1P stimulation (specifically S1P1 and S1P3) enhanced astrocytes proliferation and leads to the release of nitric oxide and the translocation of nuclear factor-κB (NFκB) within astrocytes.” Why do you use different tenses within the sentence (“enhanced” and “leads”)?

Author Response

We thank the Reviewer for his/her thoughtful comments and suggestions. Below each comment and suggestion raised in the manuscript is addressed.

Reviewer 1

  • Pg 2, line 49. The first presented case with the “inguinal hypoesthesia and left lower limb hyposthenia” could also be caused by a new spinal lesion. Why didn’t you perform a spinal MRI in addition to the cerebral MRI?

We thank the reviewer for this comment.

We apologize for the inaccuracy. The patient performed a re- baseline MRI 6 months after treatment initiation (brain, cervical and dorsal spinal cord) that was found stable compared to the previous scan (we have added this information in the manuscript). Since the patient was lost to follow-up, we could not perform a new MRI, including the lumbo-sacral tract of the spinal cord.

  • Pg 1, line 48; pg 2, line 59; pg 2, line 75. In all three cases you describe a “treatment failure” under IFN, please be more precise and describe, how treatment failed in each one of the patients.

We thank the reviewer for this comment. We have now defined treatment failure for each patient.

  • In all three cases you do not describe the duration between stopping fingolimod and starting siponimod. In the discussion you mention that there was no wash-out period, so you stopped fingolimod on one day and started siponimod on the next day?

We thank the reviewer for this comment. In all three cases, we switched from fingolimod to siponimod without a wash-out period while strictly monitoring the blood counts as outlined in technical guidelines. We have added this information in each corresponding paragraph to clarify the manuscript.

  • It is important to mention that the term “rebound” has not been precisely defined in the literature. Most commonly it describes a level of disease activity after stopping of an immunotherapy which is higher than before initiation of this immunotherapy. Moreover, the terms disease reactivation and disease rebound should not be used interchangeably.

We thank the reviewer for allowing us to clarify the difference between disease rebound and reactivation. We now clarified this issue in the introduction.

We also specified (line 98-100):

“In our patients, we did not observe a higher disease activity after switching to siponimod than before fingolimod initiation. Therefore, a diagnosis of disease reactivation rather than rebound was proposed.”

Moreover, we rectified in the different sections of the manuscript the parts that might be somewhat confusing concerning rebound/disease reactivation.

  • In the discussion, you fail to mention other differences between fingolimod and siponimod besides the receptor selectivity, such as partial agonism/antagonism on the S1PR, precise targeting of the S1PR and different pharmacokinetic characteristics, which presumably contribute to the differences between the various S1P receptor modulators (see for example “FTY720 (fingolimod) for relapsing multiple sclerosis,” Expert Rev Neurother, 2008, “Sphingosine 1-Phosphate Receptor Modulators for the Treatment of Multiple Sclerosis,” Neurotherapeutics, 2017, and “Sphingosine 1-phosphate Receptor Modulator Therapy for Multiple Sclerosis: Differential Downstream Receptor Signalling and Clinical Profile Effects,” Drugs, 2021).

We thank the reviewer for this suggestion. Now we have mentioned the differences between fingolimod and siponimod as suggested. (Lines 147-179).

Minor aspects

  • Page 1, line 34: “A 47-year-old woman was diagnosed with RRMS at the age of 23 years old”. Please omit the word “old”, it is not necessary.

We amended it.

  • Page 1, lines 37-38: “… although an increase in brain and spinal cord MRI T2-lesion have been observed.” Please change “have” to “has”.

We amended it.

  • Page 1, lines 38 and 40: “Her EDSS progressed from 2,5 to 4.” “Since 2014 her disability gradually worsened, reaching in 2021 an EDSS of 6.” Those sentences are a bit contradictory, please adapt.

We clarified it.

  • Page 2, line 44: “left lower limb hyposthenia”: I would advise to change “hyposthenia” to “weakness” throughout the manuscript.

We corrected it.

  • Page 2, line 53: “A 36-year-old man was diagnosed with RRMS at the age of 23 years old.” Please omit the word “old”.

We amended it.

  • Page 2, lines 55-56: “He was disease activity (relapse, new T2 lesion and/or Gd enhancing lesion) free from 2011 to 2018…” Please change the sentence order to “he was free from disease activity (relapse, new T2 lesion and/or Gd enhancing lesion) from 2011 to 2018”. Moreover, you omit in this definition of disease activity the enlarging T2 lesions, which is a bit unusual. Were the enlarging T2 lesions not included in your definition of disease activity?

The enlarging T2 lesions were always included in our definition. We specified it.

  • Page 2, lines 74-75: “In 2017, she was switched again to fingolimod for the presence of high JCV index (3.2).” To be precise this sentence is not entirely correct. She was not switched due to the high JCV index, but the associated PML risk. Moreover, please introduce all abbreviations in your manuscript, when they are first used. 

We amended it.

  • Page 3, lines 101-102: “. Three patients out of 13 presented disease reactivations (20%)”. According to my calculations 3 out of 13 equals 23%.

We amended it.

  • Page 3, lines 116-119: “In support of a central immunomodulatory role exerted by fingolimod, studies in EAE showed that S1P stimulation (specifically S1P1 and S1P3) enhanced astrocytes proliferation and leads to the release of nitric oxide and the translocation of nuclear factor-κB (NFκB) within astrocytes.” Why do you use different tenses within the sentence (“enhanced” and “leads”)?

We apologize for the inaccuracy. We amended it.

Reviewer 2 Report

Dears,

Manuscript entitled "Disease reactivation in Secondary Progressive Multiple Sclerosis patients switching from fingolimod to siponimod: a case series" presents 3 cases of patients with multiple sclerosis (MS) in whom exacerbation occurred after switching treatment from fingolimod to siponimod. It is an important possible adverse event in MS patients as siponimod has been recently available for secondary progressive MS patients. There is only one publication regarding similar case. 

The manuscript requires English editing as there are some mistakes (grammar, style and spelling).

In case 1 there is some inaccuracy (in line 47): there is a statement, that a patient "partially recovered from her new symptoms needing single aid to walk", however she had already had EDSS=6.0 and it means she needed one-side help in walking. 

The article could be published after corrections. 

Author Response

We thank the Reviewer for his/her thoughtful comments and suggestions. Below each comment and suggestion raised in the manuscript is addressed.

Reviewer 2

Dears,

Manuscript entitled "Disease reactivation in Secondary Progressive Multiple Sclerosis patients switching from fingolimod to siponimod: a case series" presents 3 cases of patients with multiple sclerosis (MS) in whom exacerbation occurred after switching treatment from fingolimod to siponimod. It is an important possible adverse event in MS patients as siponimod has been recently available for secondary progressive MS patients. There is only one publication regarding similar case. 

The manuscript requires English editing as there are some mistakes (grammar, style and spelling). 

We thank the reviewer for the suggestion. We have revised the manuscript.

In case 1 there is some inaccuracy (in line 47): there is a statement, that a patient "partially recovered from her new symptoms needing single aid to walk", however she had already had EDSS=6.0 and it means she needed one-side help in walking. 

We apologize for the inaccuracy. We have specified that during the relapse, the patient experienced a worsening in EDSS (from 6 to 6,5), requiring constant bilateral assistance to walk. After five days of pulsed steroid therapy, she partially recovered from her new symptoms, enough to need just a single aid to walk again.

The article could be published after corrections. 
